# A lesson for teaching fundamental Machine Learning concepts and skills to molecular biologists

**Rabea Müller** [1]   **Akinyemi Mandela Fasemore** [1 2 3]   **Muhammad Elhossary** [1]   **Konrad U. Förstner** [1 4]

## Abstract

Machine Learning represents an invaluable set of tools for the analysis of data in molecular biology as well as bio-medicine. Here we present a training approach to teach fundamental machine learning skills to researchers in their early career stage (PhD and postdoc level) with the aim to empower them to apply these methods in their own research projects. The content was developed for being delivered in a short and intense learning period as part of a remote systems biology workshop but can be adapted to other scenarios with a less restricted time frame.

## 1. Introduction

### 1.1. The need for machine learning skills in molecular biology research

With the rapidly growing amount of data in molecular biology the application of machine learning methods becomes increasingly useful and necessary to translate data – e.g. resulting from high-throughput methods like 2nd and 3rd generation sequencing (Schmidt & Hildebrandt, 2021) or proteomics (Wen et al., 2020) – into biological insights.

Based on this development we assume that having a basic understanding of the concepts of machine learning is beneficial for researchers in molecular biology. This not only helps to critically question available methods but also to implement self-developed machine learning based solutions to answer relevant research questions. Due to the availability of powerful yet comparatively easy to handle programming packages like scikit-learn (Buitinck et al., 2013), pytorch (Paszke et al., 2019), TensorFlow (Abadi et al., 2015) and

---
[*]Equal contribution  [1]ZB MED – Information Centre for Life Science, Cologne, Germany [2]Bundeswehr Institute of Microbiology, Munich, Germany [3]University of Würzburg, Würzburg, Germany [4]TH Köln – University of Applied Sciences, Cologne, Germany.  Correspondence to: Konrad U. Förstner <foerstner@zbmed.de>.

*Proceedings of the $2^{nd}$ Teaching in Machine Learning Workshop*, PMLR, 2021. Copyright 2021 by the author(s).

Keras (Chollet, 2015), machine learning methods has become accessible to non-expert.

### 1.2. Learning outcomes and requirements

We have designed a dense training lesson with the aim to teach fundamental knowledge of machine learning approaches as well as the their application using the Python package scikit-learn, further supporting packages and Jupyter Notebooks. All the software used for this lesson is available under open source licenses. After attending the training learners should be able to include machine learning based methods in their own research.

The lesson starts with an introduction to the distinction between supervised, unsupervised and reinforcement learning. It then focuses on supervised learning methods and the topics of data cleaning, feature selection, feature encoding, scaling, model fitting, model evaluation, model comparison, cross validation and grid search. Concepts like over- and under-fitting, curse of dimensionality, strengths and weaknesses of different approaches as well as deep learning was discussed as part of the introduction. Equipped with such a basic understanding, learners should be able to extend their knowledge depending on their specific needs.

As requirements for this course, we expected that participants posses programming skills, ideally in Python, as well as a basic understanding of matrices and vectors besides a strong background in molecular biology. Although a deeper understanding of machine learning requires solid mathematical foundations (Parr & Howard, 2018), the lesson was designed without requiring them.

### 1.3. Methodology

The training program was created as part of a system biology workshop which was taught remotely in a 5 day long interactive session and due to this the time available was very limited. The lesson was broken down into several components (see Figure 1). Following a flipped classroom approach (Akçayır & Akçayır, 2018), the learners had access to pre-recorded videos before the actual course. In this phase, theoretical foundations were delivered in a 45 minute video. The actual course started with a question-answer session of

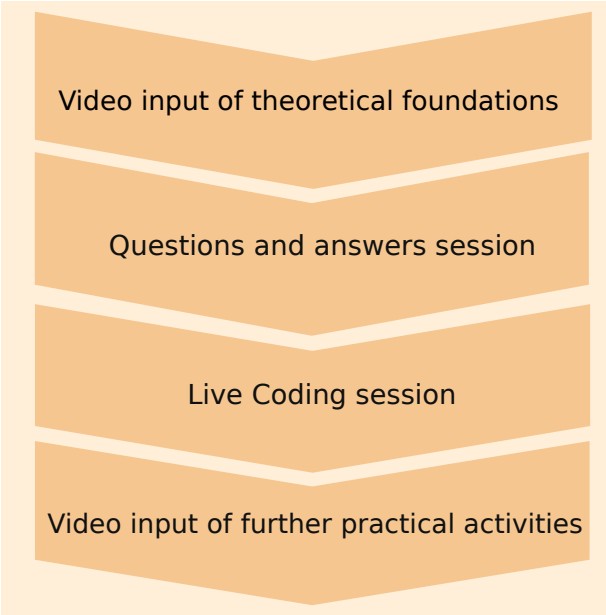

*Figure 1.* Components of the lesson

30 minutes in which open question regarding the theoretical introduction could be discussed. After that, the selected practical coding skills were taught in roughly 2 hours. In this part, the teaching methods of the "The Carpentries" (CAR; Pugachev, 2019), a community-driven organization that teaches basics of programming and data science skills to researchers and people in information-centric roles, were intensively applied. The learning methods include "Live Coding" elements, in which an instructor creates codes and executes them in front of the participants. The participants type along and replicate the code themselves instead of being taught with static code examples. The teaching methods of "The Carpentries" also include collaborative teaching and continuous feedback. Therefore, in addition to the instructor, helpers are present to take questions from the learners and to support the instructor in conveying the content. The participants can ask questions at any time and problems that occur are discussed openly with the whole group for learners. This approach converts errors into possibilities to extend comprehension and helps participants to build mental models of the content efficiently.

Due to the short duration of the actual interactive session, further contents covering further coding example are provided as videos. Learners can, depending on their interest, deepen their skills at their own pace.

## 2. Material

### 2.1. Coding environment

For teaching the implementation of machine learning solutions in Python with a Live Coding approach, we have chosen Jupyter Notebooks (Kluyver et al., 2016) as coding environment that follows the "literate programming" paradigm (Knuth, 1984). As commonly done as part of the teaching methodology of "The Carpentry", the learners start in an empty notebook which is successively extended under guidance of the instructor. For each piece of code added, explanation are given and references to the theoretical foundation are made.

### 2.2. Packages

In order to empower the learners to implement their own machine learning based solution efficiently we have chosen commonly used open source Python libraries:

The scikit-learn package (Buitinck et al., 2013) was selected as framework to teach machine learning methods due to its simplicity, strong community support and unified application programming interface (API) for classification and regression methods. Furthermore, it provides example data sets as well as numerous pre-processing and evaluation methods.

Pandas (Wes McKinney, 2010) is the Swiss army-knife in the field of data science for reading, manipulating, writing and visualisation (tabular) data. It provides the powerful "DataFrame" data structure and by that helps to access data in a format that can be processed by scikit-learn.

NumPy (Harris et al., 2020) is a foundational library for storing multi-dimensional arrays and matrices in Python and offers method to apply mathematical operation on them.

The Biopython package (Cock et al., 2009) is a collection of several tools for computational biology. Among other features it provides solution to parse numerous format including FASTA.

As we covered the classification of proteins in the lesson, we included the PyBioMed package (Dong et al., 2018), which includes several methods to encode amino acids sequences into numerical values.

### 2.3. Data sets

For selecting example data sets we applied several criteria: The data set must be open, of moderate size to ensure quick processing during the course, well documented and biologically relevant. This limited our options significantly, nevertheless we found adequate data sets which met our criteria (see Table 1):

*Table 1.* Description of used data sets.

| DATA SET NAME | SOURCE |
| --- | --- |
| BREAST CANCER | SKLEARN |
| DIABETES | SKLEARN |
| (NON-)RNA BINDING PROTEIN | ZHANG & LIU, 2016 |
| CANCER CELL EXPRESSION | WANG *et al*, 2014 |

The Breast Cancer data set is a standard machine learning data set available at the UCI Machine Learning Repository (Dua & Graff, 2017) and scikit-learn offers a function to access the data (as "Bunch" object). Aside from meeting our main criteria, it was also selected for its simplicity. The data set contains 569 data points with 30 numerical attributes. It is labelled as two classes – malignant and benign – and the classes are already in the target vector. In this lesson it served as the first example for a classification problem.

The Diabetes Data Set is another widely used data set provided by the UCI Machine Learning Repository and is also included in the scikit-learn toy data set collection. It has 442 data points with 20 (mostly numerical) attributes and serves to teach regression in the lesson.

To demonstrate how classification can be performed on protein sequences, we combined a set of RNA Binding Protein data set (2,780 sequences from 638 species) and non-RNA Binding Protein data set (7,093 sequences from 1,587 species) which were original obtained by Zhang & Liu from UniProt or the Protein Data Bank, respectively.

Furthermore, we included a cancer expression data set which originated from The Cancer Genome Atlas (TCG) and was compiled by Wang et al. for testing a Similarity network fusion (SNF) aggregation method. The data set was modified and pre-processed by us to teach a multi-class classification method for four cancer types (breast, colon, glioblastoma multiforme (GBM) and lung). The resulting set contained 518 data points (samples) with 11,925 attributes (gene expression levels).

### 2.4. Availability

All materials including the presentation used to deliver the theoretical part and the Jupyter Notebooks are Open Education Resources (OER) and available under a Creative Commons Attribution License (CC-BY). The material can be found at https://doi.org/10.5281/zenodo.5218744.

### 3. Conclusion

Machine learning has become an essential tool for the analysis of data in molecular biology. We have created a compact, multi-modal course for teaching fundamental theoretical knowledge and practical skills with the aim to enable re-searchers to include machine learning based solution into their research.

From our experiences the strategy to use a flipped classroom approach, in which the theoretical foundations are delivered as videos and the practical skills are taught in a remote live session, turned out to be successful as shown by the user feedback. As part of the overall workshop evaluation the participants were asked to rate the session and of 17 participants 0 (0%) rated the ML session as "poor", 0 (0%) as "satisfactory", 1 (5.88%) as "good", 5 (29.41%) as "very good" and 11 (64.71%) as "excellent". Those were the top results of all sessions in the whole workshop.

Still, we see several issues and possibilities to improve the lesson. The largest challenges were to deliver the core concepts of machine learning as well as practical skills in a very limited amount of time (roughly 4 h in total). We would recommend planning at least a full day of roughly 8 h to give the learners also the space to better develop their skills by working on own data sets or further given sets (e.g. in group of 3 - 4 people). We would claim that the remote learning experience and an in person mode do not differ much in terms of efficiency to deliver the content. The key principle in both modes is to keep the learners' attention by individual, direct communication through frequently asking for quick feedback and keeping them reminded that their questions can be asked any time.

In our opinion the choice of the data sets is crucial to motivate and engage the learners. While countless general introductions to machine learning exist in numerous forms, we see a need for domain specific content that empowers the learning quickly to apply machine learning methods in their research field.

As the overall experience was positive, we consider converting the developed material into a lesson as part of the incubator of "The Carpentries" (inc) to increase its visibility and lay the foundation for a sustainable extension of it.

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
