# OpenReview forum: "A lesson for teaching fundamental Machine Learning concepts and skills to molecular biologists"
_ecmlpkdd.org/ECMLPKDD/2021/Workshop/TeachML — TeachML 2021_

### Official Review · Reviewer_a32U · 2021-07-08
**Well thought out introductory course with innovative teaching strategies**

**Rating:** 8
**Confidence:** 4

**Review:**

The authors present their approach for an introductory one week course on ML for biologists.
I think the course is well designed and I especially liked the focus on open source software and datasets.
Also, it is nice that all aspects of the data lifecycle seem to get covered. However, maybe this is a lot to digest in a relatively short amount of time.

The inverted classroom approach is also a good idea that probably saves a lot of time since the focus can be on things that were not well understood by the participants. I wish the authors could have shared their experiences (and learner feedback) using this approach and contrast it with a more traditional teaching style.
Moreover, it seems that the authors took great care to convey both, theory and practice, and chose a live coding approach that allows turning errors into opportunities for learning!

Things that could have been improved are the following:
Since the course was explicitely designed as a remote event, I wish that the authors had discussed some teaching strategies that might be different than when teaching in-person , since I think that the remote nature of things may shift the teaching focus (students might get distracted more easily, the way questions can be raised, how to check if everybody keeps up, dealing with connection problems, ...).

Also, here and there I found language problems or typos (especially the trailing 's' of plural forms is often omitted)

Additionally, maybe at least one image dataset could also be included in future material to introduce ML capabilities beyond tabular data to the audience (which I think especially relevant for biologists)

---

### Official Review · Reviewer_AXoe · 2021-07-16
**Great description of a useful workshop, but could use more reflection**

**Rating:** 8
**Confidence:** 4

**Review:**

This paper describes a 5-day long interactive workshop developed to teach machine learning concepts to molecular biologists.

Having a bioinformatics background, I really appreciate the niche that this workshop fills, and I think that the talk would be a valuable addition to the workshop to hear about how AI/ML is taught to domain experts.

Pros:
* The authors did a great job of making the case for why this workshop was useful,
* Explained desired learning outcomes
* Very thorough in describing coding environment and datasets used (and why)

Cons:
* I wish the authors had expanded the Conclusion section to talk more about the feedback they received, and lessons learned
* Figure 1 didn't feel like it added much to the paper.

---

### Decision · Program_Chairs · 2021-07-21

**Decision:**

Accept

**Comment:**

Congratulations! The reviewers agree that this paper should be accepted.

Camera-ready version is due August 18, 2021. As you prepare the camera ready version, please take the reviewers comments into consideration.

We look forward to your participation at the workshop on September 13, 2021. We invite you also to join us for the satellite event on September 08, 2021. Schedules for both the workshop and the satellite event will be forthcoming.